# Forecasting the Pharmacological Mechanisms of *Plumbago zeylanica* and *Solanum xanthocarpum* in Diabetic Retinopathy Treatment: A Network Pharmacology, Molecular Docking, and Molecular Dynamics Simulation Study

**DOI:** 10.3390/biology13090732

**Published:** 2024-09-18

**Authors:** Nilanchala Sahu, Rama Tyagi, Neeraj Kumar, Mohd. Mujeeb, Ali Akhtar, Perwez Alam, Swati Madan

**Affiliations:** 1Sharda School of Pharmacy, Sharda University, Greater Noida 201310, Uttar Pradesh, India; nilanchalasahu24@gmail.com; 2Galgotias College of Pharmacy, Greater Noida 201310, Uttar Pradesh, India; tyagirama8@gmail.com; 3Department of Pharmacognosy & Phytochemistry, School of Pharmaceutical Education and Research, Jamia Hamdard, M. B. Road, New Delhi 110062, India; neerajpaswan225@gmail.com (N.K.); mmujeeb@jamiahamdard.ac.in (M.M.); 4Department of Pharmacognosy, College of Pharmacy, King Saud University, P.O. Box 2457, Riyadh 11451, Saudi Arabia; aakhtar@ksu.edu.sa; 5Amity Institute of Pharmacy, Amity University, Noida 201303, Uttar Pradesh, India

**Keywords:** *Solanum xanthocarpum*, *Plumbago zeylanica*, network pharmacology, molecular docking, molecular dynamics simulation

## Abstract

**Simple Summary:**

Diabetic retinopathy is a severe complication of diabetes that can lead to vision loss due to the abnormal growth of blood vessels in the retina. This study investigates how two traditional medicinal plants, *Solanum xanthocarpum* and *Plumbago zeylanica*, might be used to treat this condition. We focused on identifying the active compounds in these plants and understanding how they interact with proteins related to diabetic retinopathy. Using advanced computer simulations, we found that these plant compounds effectively bind to and influence key proteins involved in the progression of the disease. Specifically, our research highlighted that compounds from *Solanum xanthocarpum* and *Plumbago zeylanica* could target several crucial pathways and proteins associated with diabetic retinopathy. These findings suggest that these plant-derived compounds have significant potential as therapeutic agents for diabetic retinopathy. By offering a new approach to treatment, this research could help improve the vision and overall quality of life of individuals suffering from this challenging condition.

**Abstract:**

(1) Background: Diabetic retinopathy (DR) is a major complication of diabetes, marked by abnormal angiogenesis, microaneurysms, and retinal hemorrhages. Traditional Ayurvedic medicine advocates multi-target strategies for DR management. However, the mechanisms by which *Solanum xanthocarpum* (SX) and *Plumbago zeylanica* (PZ) exert therapeutic effects are not well understood; (2) Methods: To investigate these mechanisms, we employed network pharmacology and molecular docking techniques. Phytochemicals from SX and PZ were identified using the IMPPAT database and Swiss Target Prediction tool. DR-related protein targets were sourced from the GeneCards database, and common targets were identified through Venn diagram analysis. STRING and Cytoscape were used to construct and analyze protein–protein interaction networks. Pathway enrichment was performed with Gene Ontology and KEGG databases; (3) Results: We identified 28 active phytoconstituents, targeting proteins such as EGFR, SRC, STAT3, AKT1, and HSP90AA1. Molecular docking and dynamics simulations confirmed the strong binding affinities of these compounds to their targets; (4) Conclusions: The study highlights the multi-target activity of SX and PZ, particularly in pathways related to EGFR tyrosine kinase inhibitor resistance and PI3K–AKT signaling. These findings provide valuable insights into their therapeutic potential for DR, suggesting the effective modulation of key molecular pathways involved in the disease.

## 1. Introduction

Diabetic retinopathy (DR) is a critical microvascular complication of diabetes that can lead to progressive vision loss and, ultimately, blindness if not effectively managed. As the prevalence of diabetes continues to rise globally, the incidence of DR is expected to follow suit, presenting substantial health and economic challenges [1]. The Global Burden of Disease Study has reported that DR is a leading microvascular complication associated with diabetes, with projections indicating a global prevalence of 22.27% by 2045 [2]. This trend highlights the urgent need for innovative therapeutic strategies to prevent and treat DR.

Existing treatments, such as anti-vascular endothelial growth factor therapies, laser photocoagulation, and corticosteroids, while beneficial, have limitations including adverse effects and incomplete efficacy [3]. Anti-diabetic medications are essential for managing diabetes and influencing DR progression, but they also pose challenges like hypoglycemia, weight gain, and cardiovascular risks [4]. Therefore, exploring new treatments that can effectively manage DR while minimizing adverse effects is imperative.

*Plumbago zeylanica* (PZ) and *Solanum xanthocarpum* (SX) are two medicinal plants traditionally used in Ayurvedic medicine, renowned for their diverse pharmacological properties [5,6]. The selection of SX and PZ for investigating DR is justified by their significant pharmacological effects, which are directly relevant to managing the disease. PZ is recognized for its potent anti-inflammatory and antioxidant activities. These properties are beneficial for DR, as chronic inflammation and oxidative stress play critical roles in the disease’s progression. PZ’s anti-inflammatory effects help to reduce chronic retinal inflammation, which exacerbates DR. Moreover, plumbagin, a key compound in PZ, has strong antioxidant effects that combat oxidative stress and influence angiogenesis pathways. This helps in preventing abnormal blood vessel formation, a hallmark of DR [7]. Plumbagin from PZ has been found to reduce oxidative stress and interfere with the angiogenesis pathways mediated by vascular endothelial growth factor receptor 2 [8]. SX is noted for its anti-diabetic and antioxidant properties. Its anti-diabetic effects help regulate blood glucose levels, addressing one of the primary risk factors for DR by stabilizing hyperglycemia. This is crucial for preventing the onset and progression of the disease. Additionally, SX’s antioxidant properties are essential in mitigating oxidative stress, which contributes to retinal damage in DR. Compounds such as apigenin and scopoletin within SX have demonstrated effectiveness in their anti-diabetic and antioxidant properties [9] of neutralizing free radicals, thereby protecting the retinal cells from oxidative damage and reducing the inflammation associated with DR [10]. Overall, the pharmacological effects of SX and PZ, including their roles in blood glucose regulation, oxidative stress reduction, and inflammation modulation, make them promising candidates for alternative treatments and underscore their potential in advancing DR management. Research into the bioactive compounds of these plants has shown promising results. Despite these promising findings, the precise pharmacological mechanisms of PZ and SX in DR treatment remain underexplored.

Advanced methodologies like network pharmacology, molecular docking, and molecular dynamics (MD) simulation provide powerful tools to investigate the complex interactions between bioactive compounds and their biological targets. Network pharmacology helps us understand the multi-target interactions within biological networks, offering insights into the therapeutic effects of medicinal plants. Molecular docking predicts the binding affinity and orientation of bioactive compounds with specific protein targets, while MD simulation provides a dynamic view of these interactions, offering deeper insights into the stability and behavior of drug–target complexes.

This study aims to explore the pharmacological mechanisms of PZ and SX in DR treatment using an integrated approach combining network pharmacology, molecular docking, and MD simulation. By identifying the key bioactive compounds, the potential molecular targets, and the underlying pathways, this research seeks to elucidate the therapeutic potential of these medicinal plants in DR management. The findings could contribute to developing novel, multi-targeted therapeutic strategies for DR, integrating traditional medicinal knowledge with modern pharmacological techniques.

## 2. Materials and Methods

### 2.1. Active Phytoconstituents of SX and PZ and Their Related Target Screening

The Indian Medicinal Plants, Phytochemistry, and Therapeutics 2.0 (IMPPAT, https://cb.imsc.res.in/imppat/home, accessed on 11 April 2024) database was searched using the terms “*Solanum virginianum/Solanum xanthocarpum*” and “*Plumbago zeylanica*” to identify the chemicals in these plants [11]. The bioavailability score (BS) determines a drug’s absorption into the bloodstream, while drug-likeness (DL) reflects how closely a compound resembles an approved drug in terms of structure and biology. Based compounds with a BS of ≥0.30 [12] and DL of ≥0.18 [13] were selected based on Lipinski’s rule-of-five and quantitative drug-likeness criteria. Swiss Target Prediction (www.swisstargetprediction.ch, accessed on 12 April 2024) [14] was used to retrieve targets, with the gene symbols and information sourced from UniProt (https://www.uniprot.org, accessed on 13 April 2024) [15].

### 2.2. Network Construction of Active Phytoconstituents and Related Targets

Cytoscape 3.10.2 (https://www.cytoscape.org, accessed on 14 April 2024) is free software used to organize, analyze, and construct networks from imported data with flexibility. Using this tool, a network was built to predict the interaction between proteins and active phytoconstituents [16].

### 2.3. Collection of Potential DR-Associated Targets

Collecting disease-related genes is crucial for understanding the mechanisms of herbs and their active phytoconstituents. DR-associated targets were obtained from the GeneCards database (https://genecards.org, accessed on 15 April 2024) using the keyword “diabetic retinopathy” [17].

### 2.4. Screening Phytoconstituent-Disease Overlapping Targets

The screened data of the SX and PZ phytoconstituent-related targets and the targets for DR were imported to Venny 2.1.0 (https://bioinfogp.cnb.csic.es/tools/venny/, accessed on 15 April 2024) to construct the Venn diagram [18]. This Venn diagram expressed the intersection of the screened targets of SX and PZ and the disease DR.

### 2.5. Network Construction of Phytoconstituent–Disease Common Targets

The STRING tool (https://string-db.org/, accessed on 15 April 2024) [19] was used to generate a protein–protein interaction (PPI) network, showing functional connections between the proteins. The PPI setup included selecting Homo sapiens proteins, confidence-based network edges, experimental interaction sources, and a minimum interaction score of 0.400 (medium confidence) [20]. Data from STRING was imported into Cytoscape 3.10.2 for further analysis [21]. The CytoHubba plug-in, using the Degree method, ranked the top ten target proteins with the shortest paths [22]. Molecular docking and the phytoconstituents–targets–pathways network further guided the selection of key targets.

### 2.6. Gene Ontology (GO) and Kyoto Encyclopedia of Genes and Genomes (KEGG) Pathway Enrichment Analysis

In a computational analysis, the GO and KEGG pathways were used to precisely characterize core biological activities in the human cells [23]. Enrichment analysis was performed using the bioinformatics platform (http://www.bioinformatics.com.cn, accessed on 16 April 2024) [24]. The GO analysis categorized genes into biological processes (BPs), cellular components (CCs), and molecular functions (MFs). KEGG enrichment highlighted significantly enriched BPs, CCs, and MFs with a *p*-value <0.05 and an enrichment factor >1.5 [25]. The results of GO and KEGG enrichment analyses were visualized as bubble charts.

### 2.7. Construction of PZ and SX Phytoconstituents–Targets–Pathways Network

The software “Cytoscape version 3.10.2” was used to create a network of phytoconstituents, targets, and pathways that described the mechanism of action of SX and PZ for DR. The nodes and edges, which can be represented as phytoconstituents, targets, or disease-related routes and links, were present in a network. 

### 2.8. Molecular Docking 

Molecular docking was performed using Schrödinger^®^ (Maestro, Version 12.8, 2021-2, Windows-x64) [26]. Docking was executed with extra precision (XP), and the proteins were prepared with restrained minimization using the OPLS4 force field. Grid sites were generated using the Glide^®^ receptor grid generator, with a 20 Å docking box centered on the target protein’s active residues. Phytoconstituent 3D structures were obtained from PubChem (https://pubchem.ncbi.nlm.nih.gov, accessed on 18 April 2024) and prepared with OPLS4 for a pH of 7.0 ± 2.0 [27]. The propensity for spontaneous binding between the receptor and ligand was assessed based on docking scores, with more negative values indicating better binding affinity. A docking score below −5 kcal/mol was considered indicative of excellent binding activity [28]. Visualization of the results, including 3D and 2D diagrams of the docked target–ligand complexes, was carried out using PyMOL 3.0 (https://pymol.org/, accessed on 26 April 2024) [29] and Discovery Studio BIOVIA 2024 (https://discover.3ds.com/, accessed on 26 April 2024) [30]. The top docking results, with the compounds from SX and PZ binding to the key potential targets, were selected for subsequent MD simulations and compared with the reference ligands bound to the receptors.
Ligand Efficiency=GlideScore/N
where N is the number of heavy (non-hydrogen) atoms in the ligand. 

### 2.9. MD Simulation 

To assess the stability of the most promising docked complexes, MD simulations were conducted using Gromacs 2020.4 [31], running on a workstation equipped with an Intel Xeon E3-1245 processor (8 cores at 3.50 GHz), 32 GB RAM, and an NVIDIA Quadro P5000 GPU [32]. The protein topology was generated using the CHARMM36 all-atom force field [33], and solvation was accomplished with TIP3P water molecules [34] via the ‘gmx solvate’ command in GROMACS. Ligand topology was created using the CHARMM General Force Field (CGenFF) [35]. The MD simulation was performed within a dodecahedron box, ensuring that the protein–ligand complex was centrally located with at least a 10 Å distance from the box boundaries. Sodium chloride (NaCl) was added to a concentration of 150 mM to replicate physiological conditions [36,37]. The parameters for Na^+^ were as follows: charge = +1.0 e, Lennard-Jones parameters: σ = 2.58 Å, and ε = 0.4184 kJ/mol. For Cl^−^, the parameters were as follows: charge = −1.0 e, Lennard–Jones parameters: σ = 4.40 Å, and ε = 0.4184 kJ/mol [38]. The system underwent energy minimization using the steepest descent method, with a maximum of 50,000 steps. Following minimization, isothermal–isochoric (NVT) and isothermal–isobaric (NPT) ensembles were simulated at 300 K and 1.0 bar pressure, respectively, using the Berendsen thermostat [39] and the Parrinello–Rahman barostat [40]. A production run of 200 ns was executed after equilibration, employing a 2 fs time step with the leapfrog integrator [41]. The LINCS algorithm was used to constrain the bond lengths during the NVT, NPT, and production phases [42]. The MD simulation results were analyzed for the root mean square deviation (RMSD), root mean square fluctuation (RMSF), radius of gyration (Rg), and solvent accessible surface area (SASA). Each simulation was independently conducted in triplicate, and the results were presented as mean values with standard errors.

### 2.10. Free Energy Calculation (MM-GBSA)

The free energy of protein–ligand complex formation was determined using the Molecular Mechanics/Generalized Born Surface Area (MM/GBSA) method, as previously described [43]. First, the docked complexes were optimized using the Molecular Mechanics (MM) approach. Next, energy minimization was carried out with the OPLS4 force field in conjunction with the Generalized Born Surface Area (GBSA) continuum solvent model. The binding free energies of the protein–ligand complexes were then calculated using the following equations:$$∆G_{Bind}=∆E_{MM}+∆G_{Solv\_GB}+∆G_{SA}$$
where ∆G_Bind_, ∆E_MM_, ∆G_(Solv_GB)_, ∆G_SA_ were the binding free energy between a ligand and its target, molecular mechanical energy, solvation free energy calculated using the Generalized Born (GB) model, non-polar solvation free energy, respectively.
$$∆E_{MM}=E_{complex}-\left(E_{protein}+E_{ligand}\right)$$
where *E*_Ligand_, *E*_Protein_, and *E*_Complex_ were the minimized energies of the ligand, protein, and protein–ligand complex, respectively.
$$∆G_{solv\_GB}=G_{solv_{GB}(complex)}-G_{solv_{GB\left(protein\right)}}+G_{solv\_GB(ligand)}$$
where G_Solv_GB_ (Ligand), G_Solv_GB_ (Protein), and G_Solv_GB_ (Complex) were the free energies of solvation of the ligand, protein, and protein–ligand complex, respectively.
$$∆G_{SA}=G_{SA(compelx)}-G_{SA\left(protein\right)}+G_{SA(ligand)}$$
where, G_SA_ (Ligand), G_SA_ (Protein), and G_SA_ (Complex), were the surface area energies of the ligand, protein, and protein–ligand complex, respectively.

The free energy, in the Prime-MM/GBSA method, is calculated as follows:$$∆G_{Bind}=∆G_{Coulomb}+∆G_{vdW}+∆G_{Covalent}+∆G_{H-bond}+∆G_{Sol\_Lipo}+∆G_{Solv\_GB}\;+∆G_{Packing}+∆G_{Self-contact}\;\;\;\;\;\;$$
where ∆G_Coulomb_, ∆G_vdW_, ∆G_Covalent_, ∆G_(H-bond)_, ∆G_(Sol_Lipo)_, ∆G_(Solv_GB)_, ∆G_Packing_, ∆G_(Self-contact)_, were Coulombic (electrostatic) contribution, van der Waals contribution, hydrogen bond contribution to the binding free energy between the ligand and its receptor, lipophilic (hydrophobic) contribution to solvation free energy, polar solvation free energy by GB model, packing free energy contribution, self-contact free energy or self-energy of a molecule. 

### 2.11. Principal Component Analysis (PCA)

The collective motion of proteins along with their respective ligands was measured by employing the PCA approach using the Bio3D package [44]. In this method, the initial step involved removing the translational and rotational movements of the protein. Next, the covariance matrix and its eigenvectors were determined by aligning the protein’s atomic coordinates with a reference structure. The symmetric matrix was then diagonalized through an orthogonal transformation, resulting in a matrix of eigenvalues. The covariance matrix (C) is computed using the following formula.
$$C_{ij}=\langle \left(x_{i}-\langle x_{i}\rangle \right)\left(x_{j}-\langle x_{j}\rangle \right)\rangle \;\;\;\;\;\;\;\;\;\;i,j=1,2,3,\ldots ..,3N$$
where N, x_i/j_ and <x_i/j_> represent the number of Cα-atom, the Cartesian coordinate of the i^th^/j^th^ Cα-atom, and time average of all the conformations, respectively.

## 3. Results

### 3.1. Active Phytoconstituents of SX and PZ 

Using the IMPPAT database, the active phytochemical constituents of SX and PZ were selected. Up to 16 compounds of SX and 12 compounds of PZ were chosen after additional research was performed based on the specific properties of thresholds of BS ≥ 0.3 and DL ≥ 0.18; Table 1 also shows the molecular weight (kcal/mol), hydrogen bond acceptor (HBA), hydrogen bond donor (HBD), total polar surface area (TPSA), gastrointestinal absorption (GIA), and Log P for each compound.

### 3.2. Phytoconstituent–Target Network Construction

To show the relationships between the SX and PZ phytoconstituents and their respective targets, a network known as the phytoconstituent–target network was created (Appendix A). A total of 28 phytoconstituents were mapped to 955 potential targets in the network, which has 401 nodes and 983 edges. The SX and PZ phytoconstituents are shown in the blue and green rounded rectangles, respectively, in the network, whereas the putative targets were depicted as pink circles. The phytoconstituents of SX as Apigenin, Beta-sitosterol, Campesterol, Cholesterol, Sitosteryl glucoside, Stigmasterol, Stigmasteryl glucoside, Cycloartenol, Soladulcamaridine, Solasodine, Scopolin, Cycloartanol, Esculin, Caffeic acid, Esculetin, and Scopoletin, relate to targets 100, 44, 58, 56, 26, 41, 29, 26, 32, 32, 34, 35, 38, 52, 61 and 61, respectively. The phytoconstituents of PZ as 3-chloroplumbagin, Chitranone, D-Fructose, D-Glucose, Droserone, Elliptinone, Isozeylanone, Maritinone, Methylnaphthazarin Plumbagin, Plumbazeylanone, and Zeylanone related to the targets 20, 14, 11, 13, 15, 10, 64, 26, 16, 9, 26, and 45, respectively. 

### 3.3. Predicting DR-Related Targets

The information was compiled by extracting relevant targets associated with DR from the GeneCards database. A total of 5181 potential DR-related target genes were identified from the GeneCards database. The results of a Venn diagram analysis, shown in Figure 1, reveal that 217 probable target genes are found among the 5181 DR genes and the 371 potential targets of the active phytoconstituents in SX and PZ after removing the duplicates.

### 3.4. Common Targets PPI Network

A total of 217 putative target genes common to SX and PZ were imported into the STRING database for analysis and the formation of a PPI network. The network construction considered only experimental interaction sources with high-confidence network edges. As shown in Appendix A, a medium confidence score of 0.400 was selected. This resulted in a network comprising 217 nodes and 364 edges, with an average node degree of 3.35. The PPI enrichment p-value was less than 1.0 × 10^−16^. The data were then imported into Cytoscape for further analysis and network visualization (Appendix A). In this graphical representation, the nodes represent the target genes, the edges indicate interactions between two potential targets, and the extent value shows the strength of these interactions. CytoHubba was used to extract the core PPI network. Using the degree method, the top 10 nodes with the shortest paths were identified. Consequently, the core targets selected, in descending order, were EGFR, SRC, STAT3, EP300, AKT1, ESR1, HSP90AB1, CTNNB1, PIK3R1, and HSP90AA1 (as shown in Figure 2). These primary targets were chosen for the phytoconstituents–targets–pathways network and for molecular docking with the main phytoconstituents of SX and PZ in the context of DR treatment. All 217 target genes identified through the PPI study were also compiled for pathway enrichment analysis.

### 3.5. GO and KEGG Enrichment Analyses

Based on the specific criteria (*p* < 0.05 and enrichment factor >1.5), the analysis focused on the top ten significantly enriched BPs, CCs, and MFs. The bubble chart illustrates the ratio of target genes to all annotated genes within each functional pathway, with the size of the dot representing the number of target genes associated with each pathway. The analysis identified 4644 enriched BPs, 371 CCs, and 242 MFs. For BPs, the three most significantly enriched terms related to key targets were GO:0006979 response to oxidative stress, GO:0043405 regulation of MAP kinase activity, and GO:0071902 positive regulation of protein serine/threonine kinase activity (Appendix A). In terms of CCs, the top three significantly enriched terms were GO:0045121 membrane raft, GO:0098857 membrane microdomain, and GO:0098589 membrane region (Appendix A). For MFs, the top three significant enrichments were GO:0004674 protein serine/threonine kinase activity, GO:0004713 protein tyrosine kinase activity, and GO:0019902 phosphatase binding (Appendix A). Understanding these mechanisms is essential for grasping how SX and PZ act in the treatment of DR. The KEGG enrichment analysis further identified the interactions between SX and PZ and the pathways relevant to DR treatment. The top ten significant signaling pathways (*p* < 0.05), based on their enrichment scores (−log10(*p*-value)), are shown in Appendix A and Table 2. The top three pathways identified were hsa01521 EGFR tyrosine kinase inhibitor resistance, hsa04066 HIF-1 signaling pathway, and hsa05205 Proteoglycans in cancer. A total of 282 pathways were discovered, suggesting that SX and PZ may help mitigate EGFR tyrosine kinase inhibitor resistance in DR treatment.

### 3.6. SX and PZ Phytoconstituents–Targets–Pathways Network Construction

Cytoscape software was used to import the data from the KEGG analysis of target pathways to construct a network graph showing the connections between phytoconstituents, targets, and pathways. The purpose of this network graph was to make it obvious which phytoconstituent interacts with which targets and which pathways each target is a member of. The SX and PZ phytoconstituents–targets–pathways network is shown in Appendix A, which includes 551 edges connecting 130 nodes (1 SX, 1 PZ, 28 phytoconstituents, 90 targets, and 10 pathways). The network analysis revealed that at least two target genes interact with various SX and PZ phytoconstituents (except Droserone from PZ). The top ten targeted genes from the core PPI network were used to generate the network in Figure 3. Additionally, five targets (EGFR, STAT3, SRC, AKT1, HSP90AA1) were identified as potentially involved in DR-related pathways. Many of these target genes were influenced by a substantial number of active phytoconstituents. The network analysis revealed various characteristics of SX and PZ phytoconstituents and their targets in the context of DR therapy.

### 3.7. Molecular Docking Simulation of Phytoconstituents and Targets

Molecular docking was used to evaluate the active phytoconstituents and targets identified in the phytoconstituents–top 10 targets–pathways network analysis, specifically targeting EGFR (5UGB), STAT3 (6NJS), SRC (2BDJ), AKT1 (3O96), and HSP90AA1 (4BQG). The absorption, distribution, metabolism, and excretion properties of the 28 selected phytoconstituents were assessed using the SwissADME web-based program (http://www.swissadme.ch/index.php, accessed on 18 April 2024). The free binding energies (in kcal/mol) of the key targets and active phytoconstituents obtained from molecular docking simulations are presented in Table 3. Visualization of the docking outcomes was performed using PyMOL 3.0 and Discovery Studio BIOVIA 2024. A lower binding energy indicates a stronger interaction between the ligand and receptor [28]. An affinity less than −4.25 kcal/mol suggests a definite interaction, less than −7.0 kcal/mol indicates strong binding activity, and less than −5.0 kcal/mol indicates good binding activity [45].

The phytoconstituents–top 10 targets–pathways network and molecular docking score data were combined to select the best ligand for the targets. The findings identified EGFR, STAT3, SRC, AKT1, and HSP90AA1 as the main therapeutic targets. Apigenin, chlorogenic acid, and stigmasterol glucoside were identified as potential active phytoconstituents of SX and PZ for treating DR. Apigenin showed better binding affinity to EGFR, STAT3, SRC, and AKT1, with docking scores of −7.648 kcal/mol, −5.383 kcal/mol, −9.127 kcal/mol, and −3.504 kcal/mol, respectively. Isozeylanone bound strongly to HSP90AA1 with a docking score of −10.126 kcal/mol, confirmed by the phytoconstituents–top 10 targets–pathways network. Additionally, Scopolin interacted strongly with EGFR (docking score: −6.588 kcal/mol), while Maritinone showed potential binding to HSP90AA1 (docking score: −9.810 kcal/mol). Sitosteryl glucoside had a docking score of −4.970 kcal/mol with STAT3, and Scopoletin had a docking score of −5.792 kcal/mol with SRC, confirming its potential binding.

Based on data from the KEGG pathway, the SX and PZ phytoconstituents–top 10 targets–pathways network, and the molecular docking results, SRC and HSP90AA1 were concluded to be key targets in disease treatment. These targets are inhibited by Apigenin and Isozeylanone, which likely function through multi-target and multi-pathway mechanisms. Consequently, these targets were selected for MD simulation analysis.

From the results, it is indicated that Apigenin can occupy the central cavity of SRC, a site also occupied by the control inhibitor AP23464 (3-[2-(2-cyclopentyl-6-{[4-(dimethylphosphoryl)phenyl]amino}-9H-purin-9-yl)ethyl]phenol) as shown in Appendix A. The docking energy of the control inhibitor AP23464 towards SRC was estimated to be −10.0 kcal/mol, whereas the docking energy for Apigenin was calculated to be −9.127 kcal/mol. The control (AP23464) interacted with SRC through two conventional hydrogen bonds with MET341:O, along with one carbon–hydrogen bond with GLU339:O. Also, the control formed one electrostatic interaction and Pi–Donor hydrogen bond with LYS295:HZ2 (Figure 4A, Appendix A). Further, the control–SRC complex was stabilized by eight hydrophobic interactions with LEU273:CD2, THR338:CD2, LEU393:CD2, VAL281 (two interactions), LEU273, ALA293, and LYS295. Additionally, the control inhibitor formed van der Waals interactions with various residues of SRC, including GLY274, GLN275, GLY310, VAL323, ILE336, GLU339, TYR340, SER342, LYS343, GLY344, SER345, ALA403, and ASP404, as shown in Figure 4A. Conversely, the Apigenin–SRC complex was stabilized by two conventional hydrogen bonds with MET341:HN and MET341:O, along with one electrostatic interaction with LYS295:NZ, and one Pi–Sulfur interaction with MET314:SD (Figure 4B, Appendix A). Moreover, Apigenin formed four Pi–Sigma hydrophobic interactions with VAL281CG1, THR338:CG2, LEU393:CD1, and LEU393:CD2, along with six Pi–Alkyl hydrophobic interactions with VAL281, ALA293 (two interactions), LEU273, VAL323, and ALA403. Furthermore, Apigenin interacted with SRC through van der Waals interactions with several residues, including GLU310, ILE336, GLU339, TYR340, GLY344, SER345, ASP404, and PHE405, as illustrated in Figure 4B.

Additionally, Isozeylanone was able to occupy the central cavity of HSP90AA1, which is the same site targeted by the control inhibitor NMS-E973 (5-(3,4-dichloro-phenoxy)-benzene-1,3-diol), as depicted in Appendix A. The docking energy for the control inhibitor NMS-E973 towards HSP90AA1 was calculated to be −8.3 kcal/mol, while Isozeylanone showed a docking energy of −10.126 kcal/mol. The control (NMS-E973) interacted with HSP90AA1 through one conventional hydrogen bond with ASN51:HD21 and one Pi–Sulfur bond with MET98:SD. Also, the control formed eight hydrophobic interactions with ALA55, MET98, LEU107 (two interactions), PHE138 (two interactions), TYR139, and TRP162 (Figure 5A, Appendix A). In addition, the control formed van der Waals interactions with the residues of HSP90AA1, such as LEU48, SER52, ILE96, GLY97, LEU103, VAL150, and THR184. Interestingly, ASP93 exhibited an unfavorable interaction with the control ligand, as shown in Figure 5A. Conversely, the Isozeylanone–HSP90AA1 complex was stabilized by two conventional hydrogen bonds with GLY97:HN, and GLY97:O, along with one Pi–Sulfur interaction with MET98:SD (Figure 5B, Appendix A). Moreover, Isozeylanone formed two Pi–Sigma hydrophobic interactions with LEU107:CD1, along with two Pi–Sigma hydrophobic interactions with PHE138, three Alkyl hydrophobic interactions (MET98, VAL150, and VAL186), and four Pi–Alkyl hydrophobic interactions with ALA55 (two interactions), MET98, and PHE138. In addition, Isozeylanone was engaged with HSP90AA1 through van der Waals interactions with residues such as ASN51, LYS58, ILE96, GLY97, ALA111, LEU103, TRP162, and THR184 (Figure 5B).

### 3.8. Analysis of MD Simulation

MD simulation is a pivotal computational method in drug discovery, offering detailed insights into the dynamic behaviors of biological macromolecules and their interactions with potential drug candidates. By exploring the structural changes, binding strengths, and conformational fluctuations of protein–ligand complexes, MD simulations enable rational drug design and virtual screening. This process accelerates the development of new therapeutics with improved efficacy and safety profiles.

#### 3.8.1. RMSD Analysis

RMSD is a crucial metric for assessing the stability of protein–ligand complexes in MD simulations. It quantifies how the structure deviates from its initial conformation over the course of the simulation [46]. 

In this study, the stability of SRC and the SRC–Apigenin complex was evaluated by examining the RMSD of Cα-atoms over a 200 ns MD simulation. For the SRC protein alone, the RMSD showed some initial fluctuations during the first 20 ns before achieving a stable state. Over the period from 20 to 200 ns, the RMSD for SRC alone ranged between 0.138 and 0.201 nm, with an average of 0.183 ± 0.006 nm (refer to Figure 6A). Conversely, the SRC–Apigenin complex displayed initial RMSD variations within the first 5 ns, which then stabilized as stable interactions were established between the protein and ligand. For this complex, the RMSD fluctuated between 0.027 and 0.094 nm from 5 to 200 ns, with an average of 0.073 ± 0.007 nm.

Similarly, the RMSD of HSP90AA1 and the HSP90AA1–Isozeylanone complex was assessed over the same 200 ns simulation. The HSP90AA1 protein alone exhibited minor RMSD fluctuations initially, eventually stabilizing. The RMSD for HSP90AA1 alone varied from 0.077 to 0.230 nm, with an average of 0.149 ± 0.005 nm (see Figure 6C). The HSP90AA1–Isozeylanone complex showed initial RMSD variations during the first 5 ns, followed by stabilization once favorable protein–ligand interactions were formed. For this complex, the RMSD ranged from 0.040 to 0.188 nm during the 5 to 200 ns period, with an average value of 0.122 ± 0.006 nm.

#### 3.8.2. RMSF Analysis

RMSF analysis is a valuable tool for understanding how the binding of ligands influences the conformational dynamics of amino acid residues within proteins [47]. 

In this research, we employed RMSF analysis to assess the side-chain conformational changes of amino acid residues in SRC and its complex with Apigenin, as well as in HSP90AA1 and its complex with Isozeylanone. The RMSF plots typically highlight regions of heightened flexibility often found in protein loops and terminal regions. For SRC, the RMSF profile of the SRC–Apigenin complex was largely similar to that of SRC alone, indicating that binding with Apigenin induces minimal conformational changes in SRC (Figure 6B). Nevertheless, minor variations were noted, likely attributable to the insertion and positioning of Apigenin within SRC’s binding site. 

Similarly, the RMSF analysis of HSP90AA1 and its complex with Isozeylanone revealed that their profiles were closely aligned (Figure 6D). This suggests that Isozeylanone binding causes only slight alterations in HSP90AA1’s conformation. Again, minor differences were observed, which could be due to Isozeylanone’s insertion and positioning within the HSP90AA1 binding site. Overall, RMSF analysis provided valuable insights into the dynamic behavior of SRC and HSP90AA1 in the presence of their respective ligands. These findings enhance our understanding of the molecular interactions between SRC and Apigenin, as well as HSP90AA1 and Isozeylanone, and offer clues regarding the potential functional implications of these interactions.

#### 3.8.3. Rg Analysis

Rg measurements are essential for evaluating critical aspects of ligand–protein interactions, such as the ligand’s positioning within the protein’s binding site and the compactness of the resulting protein–ligand complex [48]. 

In our study, we utilized Rg measurements to analyze the behavior of SRC both independently and in the presence of Apigenin. Over a 200-nanosecond timeframe, the Rg values for SRC ranged from 1.875 to 1.923 nm in the absence of Apigenin and from 1.882 to 1.944 nm in its presence (Figure 7A). The average Rg values for SRC without and with Apigenin were 1.903 ± 0.007 nm and 1.915 ± 0.008 nm, respectively. These findings suggest that Apigenin induces a slight increase in the overall compactness of the SRC protein, indicating stable packing of the SRC–Apigenin complex throughout the MD simulation. This observation underscores Apigenin’s potential role in modulating SRC’s structural dynamics, which may affect its functional activity as a ligand.

Additionally, we employed Rg measurements to examine the behavior of HSP90AA1 both independently and in the presence of Isozeylanone. Over the 200-nanosecond timeframe, the Rg values for HSP90AA1 alone ranged from 1.692 to 1.744 nm without Isozeylanone and from 1.704 to 1.755 nm with Isozeylanone (Figure 7C). The average Rg values for HSP90AA1 without and with Isozeylanone were 1.716 ± 0.004 nm and 1.728 ± 0.004 nm, respectively. These results indicate that Isozeylanone induces a slight increase in the overall compactness of the HSP90AA1 protein, suggesting stable packing of the HSP90AA1–Isozeylanone complex throughout the MD simulation. This observation highlights Isozeylanone’s potential role in modulating the structural dynamics of HSP90AA1, which may influence its functional activity as a ligand.

#### 3.8.4. SASA

Surface area measurements, referred to as SASA, are a widely used method for evaluating the following two crucial aspects: the degree of exposure of a ligand within a protein’s binding site and the interactions between the protein–ligand complex and surrounding water molecules [49]. In this study, solvent-accessible surface area (SASA) measurements were employed to evaluate the behavior of SRC with and without Apigenin. Over the 0–200 nanoseconds timeframe, the SASA values for SRC ranged from 128.4 to 137.9 nm² without Apigenin and from 128.2 to 139.1 nm² with Apigenin (Figure 7B). The mean SASA values of SRC alone and the SRC–Apigenin complex were calculated to be 133.6 ± 0.6 nm² and 134.5 ± 0.9 nm², respectively. These results suggest that the presence of Apigenin may slightly alter the surface accessibility of the SRC protein, possibly indicating changes in its binding site conformation or interactions with surrounding solvent molecules. Similarly, SASA measurements were used to evaluate the behavior of HSP90AA1 with and without Isozeylanone. Over the 0–200 nanoseconds timeframe, the SASA values for HSP90AA1 ranged from 104.9 to 116.5 nm² without Isozeylanone and from 105.9 to 120.4 nm² with Isozeylanone (Figure 7D). The mean SASA values of HSP90AA1 alone and the HSP90AA1–Isozeylanone complex were calculated to be 110.2 ± 0.6 nm² and 111.8 ± 0.9 nm², respectively. These findings suggest that the presence of Isozeylanone may slightly alter the surface accessibility of the HSP90AA1 protein, potentially indicating changes in its binding site conformation or interactions with surrounding solvent molecules.

### 3.9. Analysis of Hydrogen Bonds

During MD simulation, understanding the stability of protein–ligand complexes is crucial. One method to assess this stability involves analyzing the formation of hydrogen bonds within the protein (intramolecular) and between the protein and the ligand (intermolecular) over time. In this study, intramolecular hydrogen bonds within the SRC protein fluctuated between 150 and 190 bonds (Figure 8A), indicating the protein’s dynamic nature during the simulation. Intermolecular hydrogen bonds between SRC and Apigenin varied from 0 to 7 bonds (Figure 8B). Despite these fluctuations, the presence of consistent hydrogen bonds throughout the MD simulation suggests that the SRC–Apigenin complex remains stable over time.

Similarly, the intramolecular hydrogen bonds within the HSP90AA1 protein ranged from 139 to 179 bonds (Figure 8C), reflecting its dynamic structure. Intermolecular hydrogen bonds between HSP90AA1 and Isozeylanone fluctuated between 0 and 6 bonds (Figure 8D). Despite these variations, the consistent hydrogen bonds throughout the simulation indicate that the HSP90AA1–Isozeylanone complex is stable.

The formation of stable hydrogen bonds between the protein and the ligand is indicative of favorable interactions, essential for maintaining the structural integrity of the protein–ligand complex. These interactions may contribute to the ligand’s effectiveness in modulating the protein’s function. Therefore, the observed stability of both the SRC–Apigenin and HSP90AA1–Isozeylanone complexes implies that Apigenin and Isozeylanone could potentially act as reliable ligands for SRC and HSP90AA1, respectively, under physiological conditions.

### 3.10. Analysis of Free Energy Calculations (MM/GBSA)

To assess the binding strength of SRC with Apigenin and HSP90AA1 with Isozeylanone, the free energy, representing the protein–ligand interactions in a solvent environment, was calculated using the MM/GBSA method. The results, shown in Table 4, reveal that the HSP90AA1–Isozeylanone complex had a lower free energy (−49.85 kcal/mol) compared to the SRC–Apigenin complex (−45.58 kcal/mol). In this context, the primary contributors to the stability of the protein–ligand complexes were Coulombic energy (ΔG_Coulomb_), lipophilic energy (ΔG_SA_ or ΔG_Sol_Lipo_), and van der Waals energy (ΔG_vdW_). In contrast, the polar solvation energy (ΔG_Solv_ or ΔG_SolGB_) and covalent energy (ΔG_Covalent_) were the main forces opposing the protein–ligand interactions. These findings are consistent with molecular docking results, highlighting that Apigenin and Isozeylanone exhibit significant inhibitory potential against SRC and HSP90AA1, respectively (Table 4).

### 3.11. PCA

PCA is a widely employed technique for assessing the overall motion of target proteins both with and without their respective ligands throughout simulations. In our study, PCA was utilized to analyze the conformational sampling of SRC in the presence of Apigenin and HSP90AA1 in the presence of Isozeylanone, along the PC1–PC2, PC2–PC3, and PC1–PC3 axes projected by the Cα-atoms (Figure 9 and Figure 10). 

Each red and blue dot on the plots represent a distinct conformational state of SRC and HSP90AA1, while the red and blue clusters signify energetically favorable regions of conformational space. In the PC1–PC2 projection, the conformational subspace occupied by the SRC–Apigenin complex ranged from −15 to +15 along PC1 (14.45%) and from −20 to +10 along PC2 (12.28%) (Figure 9A). Similarly, in the PC2–PC3 projection, the conformational space occupied by the complex spanned from −20 to +10 along PC2 (12.28%) and from −12 to +10 along PC3 (7.43%) (Figure 9B). In the PC1–PC3 projection, the conformational space ranged from −15 to +15 along PC1 (14.45%) and from −12 to +10 along PC3 (7.43%) (Figure 9C). Notably, the first three eigenvalues of SRC in the presence of Apigenin collectively accounted for 34.2% of the conformational variances (Figure 9D).

For HSP90AA1 in the presence of Isozeylanone, the PC1–PC2 projection showed that the conformational subspace ranged from −20 to +15 along PC1 (44.17%) and from −13 to +10 along PC2 (8.53%) (Figure 10A). In the PC2–PC3 projection, the conformational space spanned from −13 to +10 along PC2 (8.53%) and from −8 to +11 along PC3 (6.13%) (Figure 10B). In the PC1–PC3 projection, the conformational space ranged from −20 to +15 along PC1 (44.17%) and from −8 to +11 along PC3 (6.13%) (Figure 10C). Notably, the first three eigenvalues of HSP90AA1 in the presence of Isozeylanone collectively accounted for 58.8% of the conformational variances (Figure 10D).

These observations underscore the significant contribution of these principal components to the overall structural dynamics of SRC induced by Apigenin binding and HSP90AA1 induced by Isozeylanone binding. This insight into the dynamic behavior of these protein–ligand complexes enhances our understanding of their functional implications in a biological context.

## 4. Discussion

DR, an ocular complication of diabetes, is marked by the formation of abnormal new blood vessels, multiple microaneurysms, dot-and-blot hemorrhages, venous beading, and cotton wool spots. Traditional Ayurvedic medicine has long suggested multi-target and multi-phytoconstituent strategies to prevent and manage retinopathy. Notably, phytoconstituents from SX and PZ are highly potent and frequently used for various illnesses, including retinopathy.

To elucidate the molecular mechanisms of SX and PZ in treating DR, this study employed network pharmacology and molecular docking techniques. The following 16 active phytoconstituents from SX were selected based on their BS and DL properties: Apigenin, Beta-sitosterol, Campesterol, Cholesterol, Sitosteryl glucoside, Stigmasterol, Stigmasteryl glucoside, Cycloartenol, Soladulcamaridine, Solasodine, Scopolin, Cycloartanol, Esculin, Caffeic acid, Esculetin, and Scopoletin. Twelve phytochemicals from PZ were selected: 3-chloroplumbagin, Chitranone, D-Fructose, D-Glucose, Droserone, Elliptinone, Isozeylanone, Maritinone, Methylnaphthazarin, Plumbagin, Plumbazeylanone, and Zeylanone.

The GeneCards database provided 5181 retinopathy-related targets, while Swiss Target Prediction suggested 371 potential targets for these phytoconstituents. A comparison of these datasets identified 217 putative targets unique to the interactions of SX and PZ with DR. GO and KEGG pathway analyses enriched 4644 BPs, 371 CCs, 242 MFs, and 282 pathways from these 217 targets. GO enrichment analysis revealed the involvement of SX and PZ phytoconstituents in numerous BPs, CCs, and MFs. These included responses to oxidative stress, regulation of MAP kinase activity, positive regulation of protein serine/threonine kinase activity, membrane raft, membrane microdomain, membrane region, protein serine/threonine kinase activity, protein tyrosine kinase activity, and phosphatase binding, all of which are directly related to the development of DR.

The KEGG pathway enrichment analysis highlighted the involvement of pathways such as PI3K–AKT signaling and EGFR tyrosine kinase inhibitor resistance in the progression of retinopathy. These pathways are crucial for processes like differentiation, proliferation, and inflammation observed in DR. Analysis of the phytoconstituents–targets–pathways network demonstrated that each phytoconstituent in SX and PZ could act on multiple targets and the associated pathways implicated in DR therapy (Figure 11).

Figure 12 illustrates the selection of target proteins based on the involvement of the top 10 core targets in EGFR tyrosine kinase inhibitor resistance and PI3K–AKT signaling pathways, including EGFR, STAT3, SRC, AKT1, and HSP90AA1. Molecular docking was conducted to validate the interaction between key phytoconstituents and their corresponding protein targets. The results revealed that Apigenin and Maritinone showed the strongest binding affinities for EGFR and HSP90AA1, respectively, indicating their potential roles in modulating these proteins. Isozeylanone and Scopoletin also demonstrated favorable interactions with HSP90AA1 and SRC, respectively, by binding effectively to their active sites. Additionally, Scopolin exhibited a strong docking score with EGFR, suggesting a stable interaction within its binding pocket. Pharmacological modification of EGFR, HSP90AA1, and SRC could result in the downregulation of pathways involved in inflammation, cellular proliferation, and stress response, which are key contributors to DR progression [50]. For instance, the inhibition of EGFR can reduce abnormal angiogenesis, a hallmark of DR, while the modulation of HSP90AA1 may help stabilize retinal cell homeostasis under diabetic stress conditions [51]. Similarly, the inhibition of SRC could lead to decreased vascular permeability and inflammation, both of which are central to DR pathogenesis [52]. These findings, supported by previous studies showing the efficacy of these compounds in mitigating retinal symptoms, underscore their therapeutic potential in DR treatment by targeting and modulating key proteins involved in disease progression. 

EGFR tyrosine kinase inhibitor resistance in DR is a significant challenge, driven by various mechanisms. These include secondary mutations like T790M, the activation of alternative pathways such as proto-oncogene c-Met, hepatocyte growth factor, and Anexelekto, as well as downstream pathway aberrations like KRAS mutations and the loss of phosphatase and tensin homologs. Additionally, the impairment of the apoptosis pathway mediated by EGFR TKIs, such as the deletion of BCL2-like 11, contributes to resistance. Aside from T790M, other resistance mechanisms such as avoiding EGFR signaling and lineage transformation have also been identified, complicating therapeutic evaluation and management [53]. The angiogenesis pathway and EGFR tyrosine kinase inhibitor resistance pathway share several key protein targets, including VEGFA, PI3K, AKT, MAPK, and STAT3, which are critical for endothelial cell proliferation, migration, and survival. In DR, excessive VEGFA-mediated angiogenesis leads to abnormal blood vessel growth [54]. Similarly, in EGFR TKI resistance, angiogenesis persists through alternative pathways, such as MET, SRC, and PRKCG, even when EGFR is inhibited. This redundancy allows tumors and pathological processes to bypass EGFR blockade, sustaining angiogenesis via PI3K/AKT and MAPK/ERK signaling. Our mapping of additional targets, including BCL2, PRKCA/B, and PIK3R1, highlights the complexity of angiogenesis regulation. Targeting these overlapping proteins could offer a more effective therapeutic approach to both DR and overcoming EGFR tyrosine kinase inhibitor resistance, emphasizing the need for multi-target strategies to disrupt these interrelated pathways.

Oxidative stress significantly contributes to DR and is intricately linked with the PI3K–Akt pathway. In retinal tissue, chronic hyperactivation of the PI3K–Akt pathway in response to glucose dysmetabolism exacerbates oxidative stress. Elevated glucose levels increase oxidative stress through the enhanced production of reactive oxygen species (ROS) and the activation of pro-inflammatory pathways. Specifically, this study demonstrates that high glucose concentrations lead to increased fibronectin and α(v)β(3) integrin levels in retinal endothelial cells, which promote a pro-migratory phenotype and further oxidative damage. Concurrently, reduced GLUT-1 expression impairs glucose uptake, perpetuating metabolic dysregulation. Our pathway analysis mapped numerous critical proteins involved in oxidative stress, including PRKCA, PIK3CA, HSP90AA1, VEGFA, and BCL2. The PI3K–Akt signaling cascade, through its interaction with these proteins, amplifies oxidative stress and influences key cellular processes such as apoptosis and cellular proliferation. Targeting this pathway, along with the associated oxidative stress proteins, could provide novel therapeutic strategies for managing DR [55]. Enhanced focus on these proteins within the oxidative stress pathway will advance our understanding and potential treatment of this debilitating condition.

The binding data from our study highlight that Apigenin, Maritinone, Isozeylanone, Scopoletin, and Scopolin exhibit promising interactions with various targets, suggesting potential therapeutic benefits for DR. Figure 11 illustrates the multi-target and multi-pathway mechanisms of action that SX and PZ may employ in treating DR. The molecular docking simulation results, as shown in Figure 12, further validate the findings from the phytoconstituents–top 10 targets–pathways network analysis, reinforcing the overall therapeutic potential of SX and PZ in DR management.

## 5. Conclusions

This study represents the first comprehensive analysis of the pharmacological and molecular mechanisms of SX and PZ in the treatment of retinopathy, utilizing bioinformatics tools such as network pharmacology, molecular docking, and MD simulation. We identified 28 active phytoconstituents in SX and PZ and compiled data for 371 active phytoconstituents and 5181 DR-related targets. Through PPI analysis, ten key targets were identified, with significant roles of major signaling pathways such as EGFR tyrosine kinase inhibitor resistance and PI3K–AKT signaling in the mechanism of action of SX and PZ. Network pharmacology highlighted strong correlations between these pathways and DR, indicating that SX and PZ exert an inhibitory effect on these processes. The key target proteins identified include EGFR, SRC, AKT1, HSP90AA1, and STAT3. The study underscores the multi-target and multi-pathway mechanisms of SX and PZ, which has implications for their therapeutic potential in DR. To fully elucidate the specific therapeutic mechanisms, further in vivo and in vitro experimental validation is necessary. It is important to note that this investigation primarily relied on network databases, molecular docking, and MD simulation.

## Figures and Tables

**Figure 1 biology-13-00732-f001:**
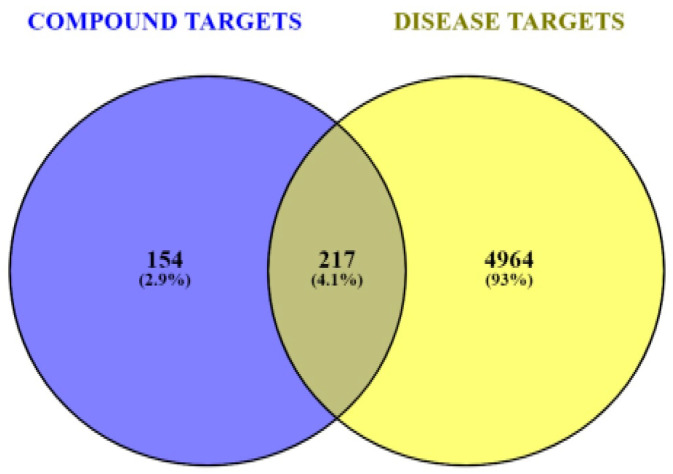
Venn diagram of screening SX- and PZ-related targets and DR-related targets.

**Figure 2 biology-13-00732-f002:**
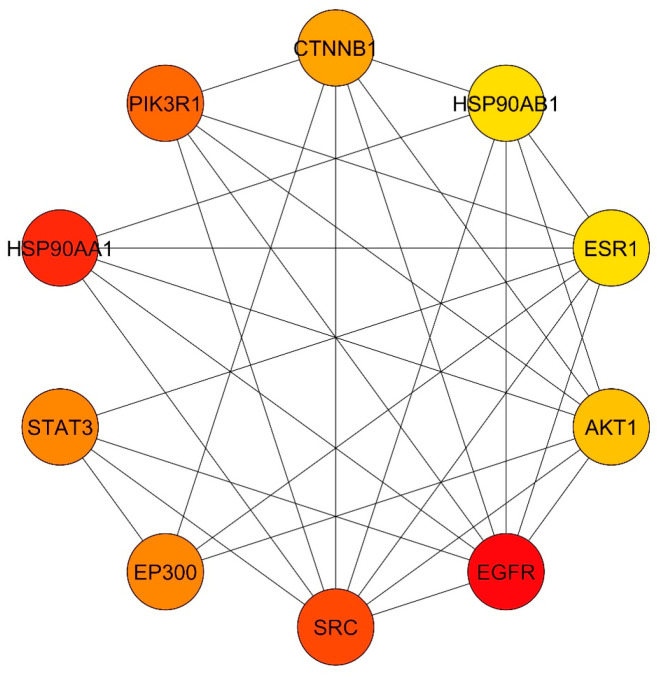
Core PPI network for top 10 targets obtained from Cytoscape–CytoHubba plugin by Degree method. Node color from red to yellow denotes descending node degree.

**Figure 3 biology-13-00732-f003:**
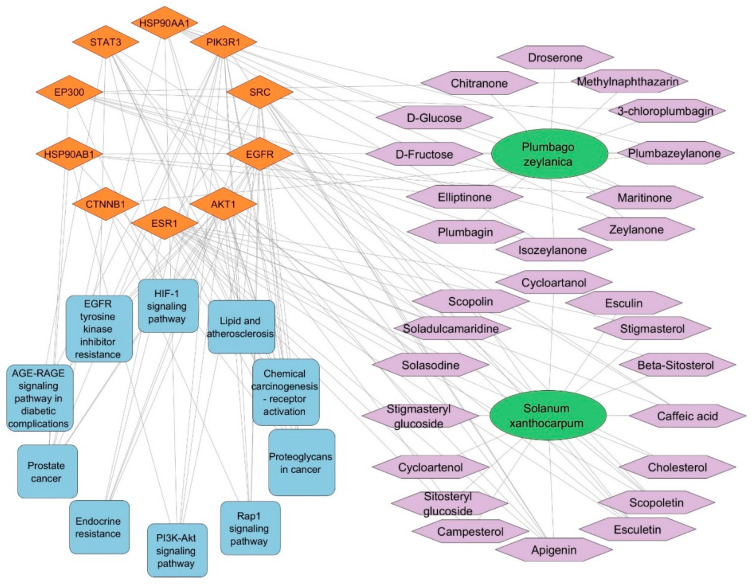
The SX and PZ phytoconstituents–top 10 targets–pathways network is illustrated as follows: Green elliptical nodes represent SX and PZ, purple hexagon nodes indicate SX and PZ phytoconstituents, orange diamond nodes denote target proteins/genes, and blue round rectangle nodes signify pathways. The edges in the network illustrate the interactions between these nodes.

**Figure 4 biology-13-00732-f004:**
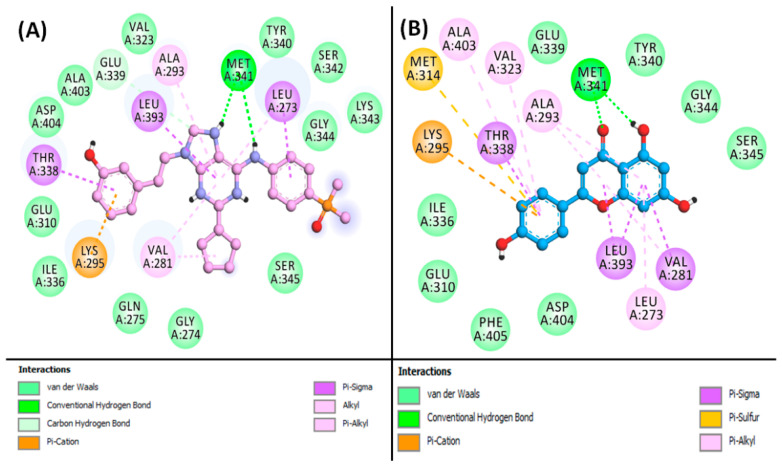
2D image for binding interaction of ligands with SRC ((**A**): AP23464, (**B**): Apigenin).

**Figure 5 biology-13-00732-f005:**
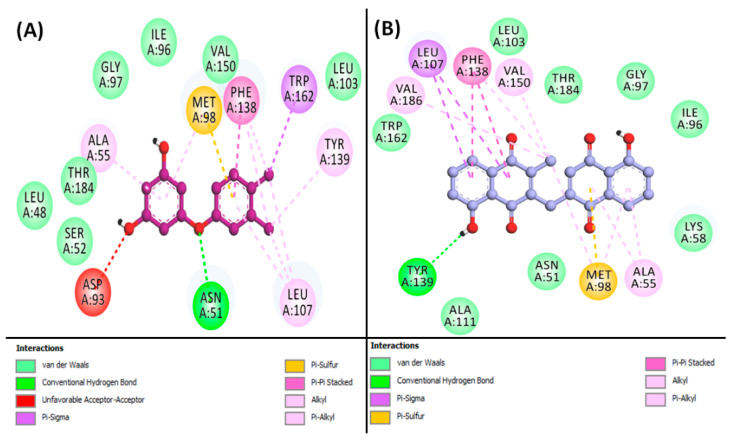
2D image for binding interaction of ligands with HSP90AA1 ((**A**): NMS-E973, (**B**): Isozeylanone).

**Figure 6 biology-13-00732-f006:**
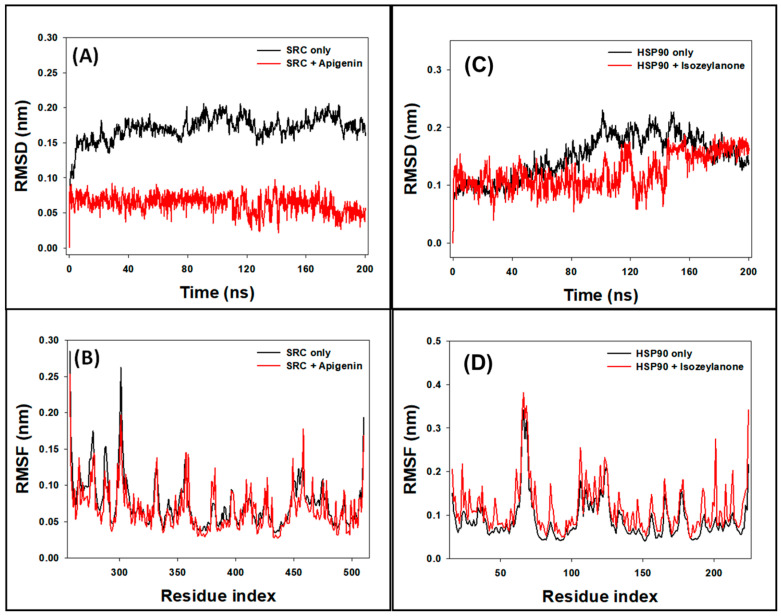
Graph of ((**A**): RMSD and (**B**): RMSF) analysis for SRC only and SRC–Apigenin complex and ((**C**): RMSD and (**D**): RMSF) analysis for HSP90AA1only and HSP90AA1–Isozeylanone complex.

**Figure 7 biology-13-00732-f007:**
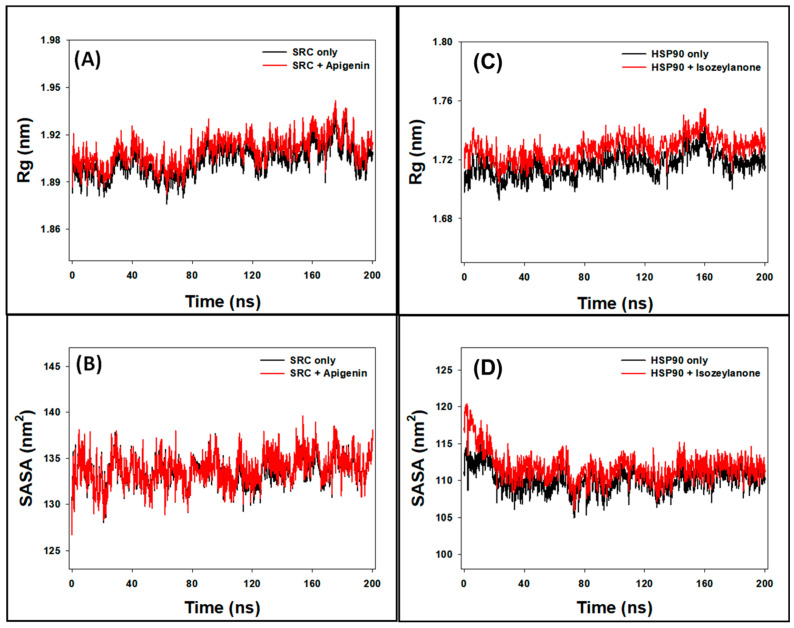
Graph of ((**A**): Rg and (**B**): SASA) analysis for SRC only and SRC–Apigenin complex and (**C**): Rg and (**D**): SASA analysis for HSP90AA1 only and HSP90AA1–Isozeylanone complex.

**Figure 8 biology-13-00732-f008:**
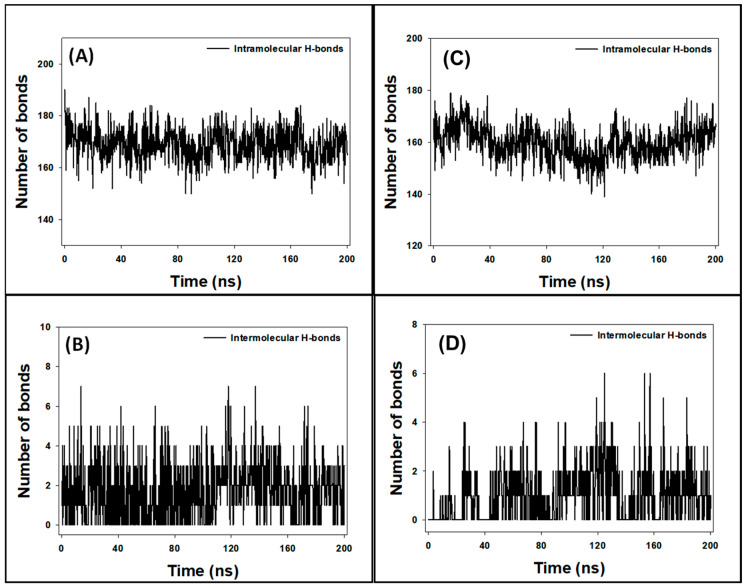
Fluctuation of intramolecular hydrogen bonds within ((**A**): SRC protein and (**C**): HSP90AA1 protein); Fluctuation of intermolecular hydrogen bonds formed between ((**B**): SRC–Apigenin complex, and (**D**): HSP90AA1–Isozeylanone complex).

**Figure 9 biology-13-00732-f009:**
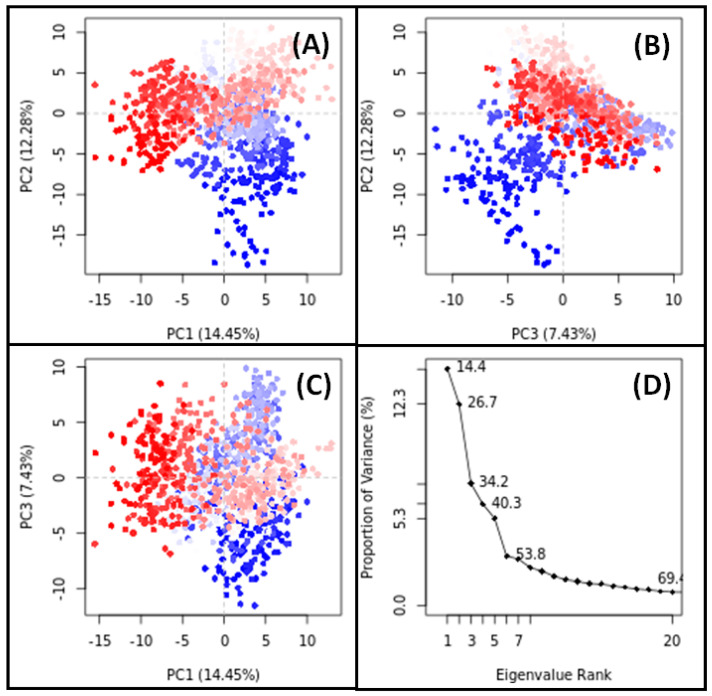
PCA results for SRC-Apigenin complex ((**A**): PC1-PC2 projection, (**B**): PC2-PC3 projection, (**C**): PC1-PC3 projection, (**D**): the conformational variances).

**Figure 10 biology-13-00732-f010:**
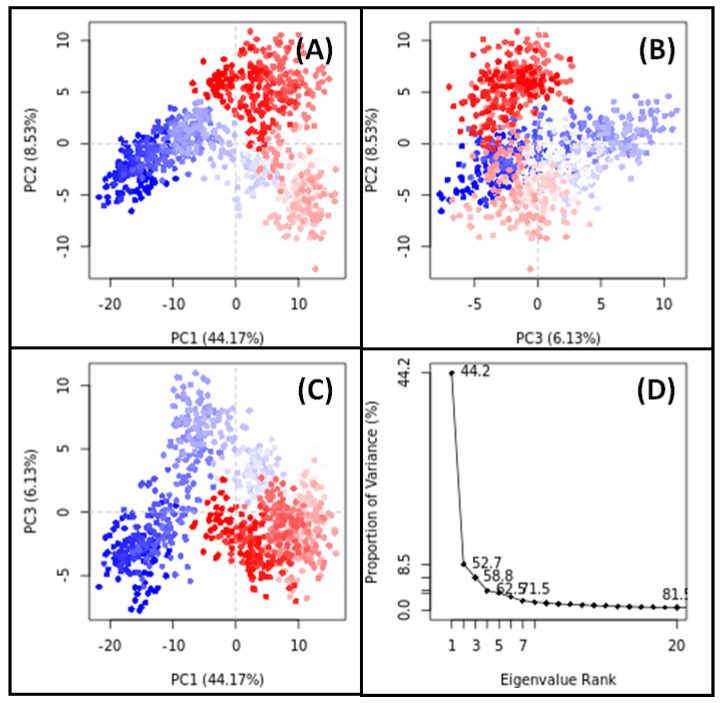
PCA results for HSP90AA1-Isozeylanone complex. ((**A**): PC1-PC2 projection, (**B**): PC2-PC3 projection, (**C**): PC1-PC3 projection, (**D**): the conformational variances).

**Figure 11 biology-13-00732-f011:**
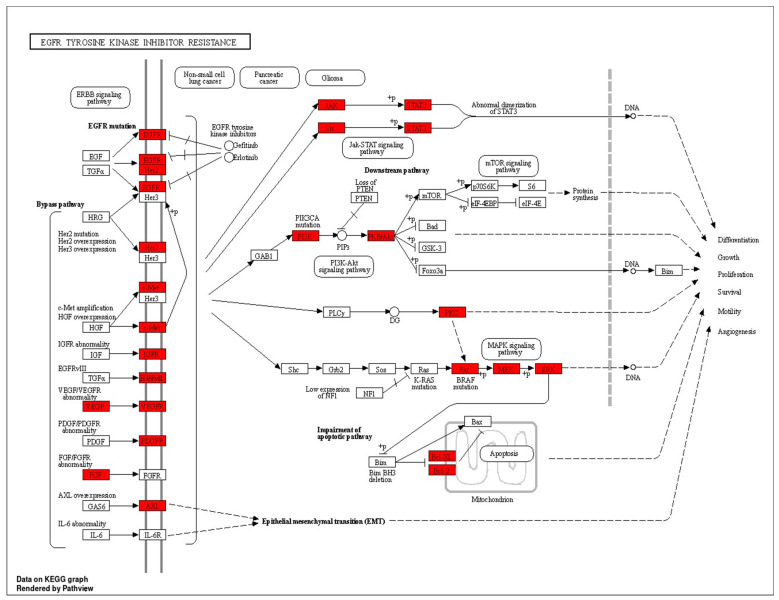
KEEG pathway diagram of EGFR tyrosine kinase inhibitor resistance (The highlighted receptors are getting targeted by the compounds).

**Figure 12 biology-13-00732-f012:**
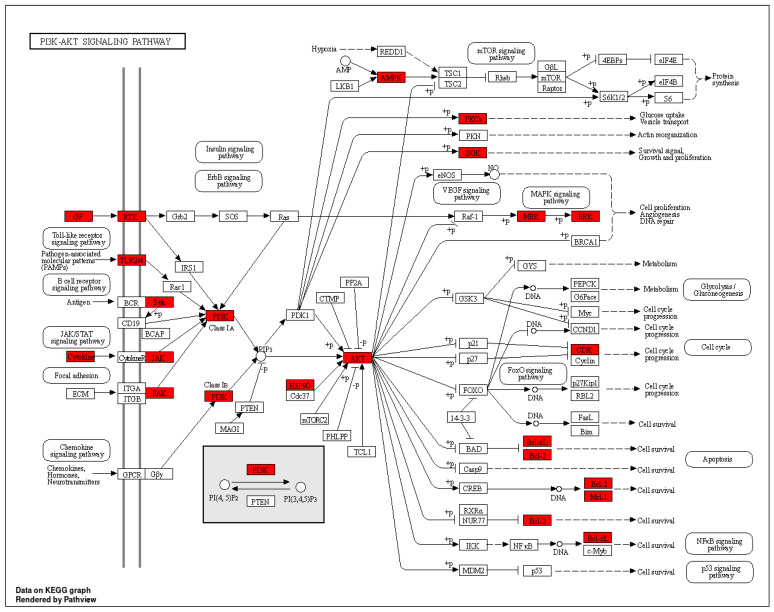
KEEG pathway diagram of PI3K–AKT signaling pathway (The highlighted receptors are getting targeted by the compounds).

**Table 1 biology-13-00732-t001:** Information of selected phytoconstituents of SX and PZ.

Imppat ID	Compound	DL	BS	Molecular Weight (kcal/mol)	Number Heavy Atoms	HBA	HBD	TPSA (Å²)	GIA	Log P
IMPHY003411	Scopolin	0.51	0.55	354.31	25	9	4	138.82	Low	1.34
IMPHY003656	Soladulcamaridine	0.53	0.55	413.64	30	3	2	41.49	High	4.15
IMPHY003952	Cycloartanol	0.47	0.55	428.73	31	1	1	20.23	Low	5.26
IMPHY004033	Solasodine	0.53	0.55	413.64	30	3	2	41.49	High	4.26
IMPHY004661	Apigenin	0.63	0.55	270.24	20	5	3	90.90	High	1.89
IMPHY005620	Esculin	0.43	0.55	340.28	24	9	5	149.82	Low	1.33
IMPHY006300	Cholesterol	0.49	0.55	386.65	28	1	1	20.23	Low	4.89
IMPHY011518	Esculetin	0.47	0.55	178.14	13	4	2	70.67	High	1.25
IMPHY011541	Scopoletin	0.7	0.55	192.17	14	4	1	59.67	High	1.86
IMPHY011642	Cycloartenol	0.45	0.55	426.72	31	1	1	20.23	Low	5.16
IMPHY011933	Caffeic acid	0.47	0.56	180.16	13	4	3	77.76	High	0.97
IMPHY012402	Campesterol	0.47	0.55	400.68	29	1	1	20.23	Low	4.97
IMPHY014836	beta-Sitosterol	0.44	0.55	414.71	30	1	1	20.23	Low	5.05
IMPHY014842	Stigmasterol	0.46	0.55	412.69	30	1	1	20.23	Low	5.08
IMPHY015071	Sitosteryl glucoside	0.26	0.55	576.85	41	6	4	99.38	Low	5.17
IMPHY015079	Stigmasteryl glucoside	0.28	0.55	574.83	41	6	4	99.38	High	5.24
IMPHY000398	Isozeylanone	0.84	0.55	374.34	28	6	2	108.74	High	2.02
IMPHY000467	Plumbazeylanone	0.37	0.55	576.55	43	9	3	163.11	Low	2.74
IMPHY001191	Plumbagin	0.67	0.55	188.18	14	3	1	54.37	High	1.79
IMPHY002828	Elliptinone	0.79	0.55	374.34	28	6	2	108.74	High	2.58
IMPHY003551	3-chloroplumbagin	0.73	0.55	222.62	15	3	1	54.37	High	1.89
IMPHY004515	Zeylanone	0.73	0.55	374.34	28	6	2	108.74	High	2.13
IMPHY004866	Droserone	0.63	0.85	204.18	15	4	2	74.60	High	1.35
IMPHY007957	Chitranone	0.79	0.55	374.34	28	6	2	108.74	High	2.58
IMPHY008637	Maritinone	0.79	0.55	374.34	28	6	2	108.74	High	2.45
IMPHY013935	Methylnaphthazarin	0.63	0.55	204.18	15	4	2	74.60	High	1.82
IMPHY014893	D-Glucose	0.29	0.55	180.16	12	6	5	110.3	Low	0.35
IMPHY014916	D-Fructose	0.29	0.55	180.16	12	6	5	110.38	Low	0.61

**Table 2 biology-13-00732-t002:** Top 10 highly enriched pathways from KEGG pathways enrichment analysis with ID, P-value, genes involved, and gene count.

ID	Pathway Name	*p*-Value	GeneIDs	Count
hsa01521	EGFR tyrosine kinase inhibitor resistance	5.3115 × 10^−23^	PRKCG/PRKCA/PRKCB/PIK3CA/VEGFA/FGF2/BRAF/BCL2/BCL2L1/EGFR/IGF1R/KDR/MET/AXL/PIK3R1/SRC/AKT1/ERBB2/MAPK1/PIK3CB/STAT3/PDGFRB/AKT2/AKT3/MAP2K1/JAK2	26
hsa04066	HIF-1 signaling pathway	2.17615 × 10^−21^	EP300/EGLN1/PRKCG/PRKCA/PRKCB/PIK3CA/VEGFA/FLT1/NOS2/BCL2/PDK1/PFKFB3/EGFR/IGF1R/PIK3R1/AKT1/TLR4/ERBB2/MAPK1/PIK3CB/STAT3/INSR/TEK/GAPDH/HK1/AKT2/AKT3/MAP2K1	28
hsa05205	Proteoglycans in cancer	2.29832 × 10^−17^	PRKCG/PRKCA/PRKCB/PIK3CA/VEGFA/FGF2/HPSE/TNF/BRAF/CTNNB1/ESR1/MMP9/MMP2/EGFR/IGF1R/KDR/MET/PIK3R1/SRC/PTK2/AKT1/PTPN6/SHH/TLR4/ERBB2/MAPK1/PIK3CB/STAT3/MAPK14/AKT2/AKT3/MAP2K1	32
hsa04151	PI3K-Akt signalling pathway	3.08738 × 10^−17^	PRKCA/PIK3CA/PRKAA1/HSP90AA1/VEGFA/FGF1/FGF2/HSP90AB1/MCL1/FLT1/SGK1/CDK2/CDK4/HSP90B1/PIK3CG/BCL2/BCL2L1/CDK6/SYK/EGFR/IGF1R/KDR/MET/PIK3R1/PTK2/AKT1/TLR4/ERBB2/MAPK1/NGFR/PIK3CB/PDGFRB/FLT4/INSR/TEK/IL2/AKT2/AKT3/MAP2K1/JAK2/CSF1R	41
hsa04015	Rap1 signaling pathway	4.77529 × 10^−17^	PRKCG/PRKCA/PRKCB/PIK3CA/VEGFA/FGF1/FGF2/FLT1/BRAF/CTNNB1/ADORA2A/EGFR/IGF1R/KDR/MET/PIK3R1/SRC/AKT1/DRD2/MAPK1/NGFR/PIK3CB/CNR1/PDGFRB/FLT4/INSR/TEK/MAPK14/AKT2/AKT3/MAP2K1/CSF1R	32
hsa05215	Prostate cancer	6.97404 × 10^−17^	EP300/PIK3CA/HSP90AA1/HSP90AB1/MMP3/BRAF/CDK2/HSP90B1/BCL2/AR/CTNNB1/MMP9/EGFR/IGF1R/PIK3R1/AKT1/ERBB2/MAPK1/PIK3CB/PDGFRB/AKT2/AKT3/MAP2K1	23
hsa04933	AGE-RAGE signalling pathway in diabetic complications	1.44583 × 10^−16^	PRKCA/PRKCB/PIK3CA/NOX4/VEGFA/MAPK8/TNF/CDK4/BCL2/PRKCD/MMP2/PIK3R1/AKT1/MAPK1/PIK3CB/F3/STAT3/MAPK14/AKT2/AKT3/JAK2/TGFBR1/MAPK9	23
hsa01522	Endocrine resistance	1.19061 × 10^−15^	PIK3CA/MAPK8/BRAF/CDK4/BCL2/ESR1/MMP9/MMP2/EGFR/IGF1R/PIK3R1/SRC/PTK2/AKT1/ERBB2/MAPK1/PIK3CB/MAPK14/AKT2/AKT3/MAP2K1/MAPK9	22
hsa05207	Chemical carcinogenesis receptor activation	4.00274 × 10^−15^	PRKCG/PRKCA/PRKCB/PIK3CA/HSP90AA1/VEGFA/FGF2/HSP90AB1/CYP1B1/HSP90B1/BCL2/AR/CHRNA7/ESR1/AHR/EGFR/PIK3R1/SRC/AKT1/VDR/MAPK1/PIK3CB/CYP1A2/CYP3A4/STAT3/PPARA/AKT2/AKT3/MAP2K1/JAK2	30
hsa05417	Lipid and atherosclerosis	5.91259 × 10^−15^	PRKCA/PIK3CA/HSP90AA1/HSP90AB1/MMP1/MMP3/MAPK8/TNF/HSP90B1/BCL2/BCL2L1/MMP9/PIK3R1/SRC/PTK2/AKT1/TLR4/MAPK1/PIK3CB/CYP2C9/NFE2L2/STAT3/PPARG/HSPA8/HSPA5/MAPK14/AKT2/AKT3/JAK2/MAPK9	30

**Table 3 biology-13-00732-t003:** Molecular docking scores, free binding energies (kcal/mol) and in the bracket glide ligand efficiency (kcal/mol per heavy atom) of key targets and active phytoconstituents.

Phytoconstituent Name (PubChem ID)	EGFR	STAT3	SRC	AKT1	HSP90AA1
Scopolin (439514)	−6.588 (−0.264)	−4.505 (−0.180)	−8.420 (−0.337)	_	−10.264 (−0.411)
Soladulcamaridine (91871142)	−3.365 ( −0.112)	−2.659 (−0.089)	−1.827 (−0.061)	_	−2.865 (−0.095)
Cycloartanol (12760132)	−2.837 (−0.092)	_	−1.455 (−0.047)	_	−1.906 (−0.061)
Solasodine (442985)	−3.704 (−0.123)	−2.061 (−0.069)	−2.306 (−0.077)	_	−2.568 (−0.086)
Apigenin (5280443)	−7.648 (−0.382)	−5.383 (−0.269)	−9.127 (−0.456)	−3.504 (−0.175)	−8.808 (−0.440)
Esculin (5281417)	−9.283 (−0.387)	−4.641 (−0.193)	−7.067 (−0.294)	−7.116 (−0.296)	−10.618 (−0.442)
Cholesterol (5997)	−3.574 (−0.128)	_	−2.079 (−0.074)	_	−4.643 (−0.166)
Esculetin (5281416)	−3.826 (−0.294)	−4.124 (−0.317)	−5.571 (−0.429)	−3.086 (−0.237)	−8.845 (−0.680)
Scopoletin (5280460)	−6.210 (−0.444)	−2.788 (−0.199)	−5.792 (−0.414)	−3.891 (−0.278)	−8.317 (−0.594)
Cycloartenol (92110)	−2.219 (−0.072)	−1.309 (−0.042)	−1.586 (−0.051)	_	−4.014 (−0.129)
Caffeic acid (689043)	−4.200 (−0.323)	−3.903 (−0.300)	−7.016 (−0.540)	−4.569 (−0.351)	−7.172 (−0.552)
Campesterol (173183)	−4.021 (−0.139)	−2.198 (−0.076)	−2.656 (−0.092)	_	−6.608 (−0.228)
Beta-Sitosterol (222284)	−3.850 (−0.128)	_	−2.872 (−0.096)	_	−6.621 (−0.221)
Stigmasterol (5280794)	−3.873 (−0.129)	_	−2.950 (−0.098)	_	−6.595 (−0.220)
Sitosteryl glucoside (70699351)	−5.998 (−0.146)	−4.970 (−0.121)	−0.067 (−0.002)	_	−3.174 (−0.077)
Stigmasteryl glucoside (70699355)	−5.316 ( −0.130)	_	_	_	_
Isozeylanone (100947536)	−5.085 ( −0.182)	−3.472 (−0.124)	−5.396 (−0.193)	_	−10.126 (−0.362)
Plumbazeylanone (100947539)	−2.564 (−0.060)	_	−3.568 (−0.083)	_	−2.814 (−0.065)
Plumbagin (10205)	−6.645 (−0.475)	−3.552 (−0.254)	−6.604 (−0.472)	−3.336 (−0.238)	−7.421 (−0.530)
Elliptinone (146680)	−5.731 (−0.205)	−3.283 (−0.117)	−6.358 (−0.227)	_	−8.926 (−0.319)
3-chloroplumbagin (338719)	−6.096 (−0.406)	−3.132 (−0.209)	−6.390 (−0.426)	_	−7.556 (−0.504)
Zeylanone (5276618)	−5.072 (−0.181)	−4.041 (−0.144)	−3.791 (−0.135)	_	−8.794 (−0.314)
Droserone (442739)	−6.278 (−0.419)	−3.252 (−0.217)	−4.828 (−0.322)	−2.235 (−0.149)	−6.633 (−0.442)
Chitranone (633072)	−5.521 (−0.197)	−1.999 (−0.071)	−8.361 (−0.299)	_	−8.287 (−0.296)
Maritinone (633024)	−5.921 (−0.211)	−3.702 (−0.1329)	−3.562 (−0.127)	_	−9.810 (−0.350)
Methylnaphthazarin (271296)	−6.589 (−0.439)	−4.281 (−0.285)	−7.300 (−0.487)	−3.210 (−0.214)	−7.406 (−0.494)
D-Glucose (5793)	−6.010 (−0.501)	−5.141 (−0.451)	−7.354 (−0.613)	−4.873 (−0.406)	−5.195 (−0.433)
D-Fructose (2723872)	−5.005 (−0.417)	−5.641 (−0.470)	−7.712 (−0.643)	−5.094 (−0.424)	−5.589 (−0.466)

**Table 4 biology-13-00732-t004:** Free energy (MM/GBSA) analysis for the interaction of SRC, and HSP90AA1 with Apigenin, and Isozeylanone, respectively.

System	Δ*G* or Δ*G*_Bind_	Δ*G*_Coulomb_	Δ*G*_Covalent_	Δ*G*_H-bond_	Δ*G*_SA_ or Δ*G*_Sol_Lipo_	Δ*G*_Solv_ or Δ*G*_SolGB_	Δ*G*_Packing_	Δ*G*_vdW_
SRC-Apigenin	−45.58	−18.22	6.91	−2.11	−15.70	19.58	−0.49	−35.55
HSP90AA1 Isozeylanone	−49.85	−31.73	2.89	−3.03	−8.09	30.21	−1.71	−38.39

## Data Availability

The data will be available from the authors upon reasonable request.

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
