# Peer review of "Forecasting the Pharmacological Mechanisms of Plumbago zeylanica and Solanum xanthocarpum in Diabetic Retinopathy Treatment: A Network Pharmacology, Molecular Docking, and Molecular Dynamics Simulation Study"

_biology, 2024, doi:10.3390/biology13090732_

Round 1

Reviewer 1 Report

Comments and Suggestions for Authors

The authors of the present study investigated the impact of 28 active compounds from Solanum xanthocarpum and Plunbago zeylanica on diabetic retinopathy using combination of network pharmacology, molecular docking, and MD simulations.

Major comments:

1.        The authors need to clarify the relationship between phytoconstituents and the protein targets explored in the manuscript.

2.        The authors need to analyze the angiogenesis pathway with more protein targets mapped.

Minor comments:

1.        The authors need to summarize key findings of the story in the abstract, the current version looks like methodology.

2.        The methodology needs to be written in very concisely.

3.        The acronyms should be expanded in the first instance.

4.        The authors need to include ligand efficiency in Table 1.

5.        The authors need to focus more on the oxidative stress pathway with more proteins targeted.

6.        The authors calculated pharmacodynamics in the manuscript, and they can also comment on pharmacokinetic properties.

7.         The authors included several main figures, a few figures can be moved to supplementary.

8.        In the introduction, the authors should write more rationale for selecting Solanum xanthocarpum (SX) and Plumbago zeylanica (PZ) plants for diabetic retinopathy (DR).

Comments on the Quality of English Language

The quality of the English language in the manuscript is acceptable.

Author Response

Dear Reviewer,

We would like to sincerely thank you for taking the time to review our manuscript submitted to Biology MDPI. We greatly appreciate your valuable comments and suggestions, which have helped us to improve the quality of our work.

We are pleased to inform you that we have addressed all of your comments thoroughly. In particular, we have provided a more comprehensive background in the introduction and included all relevant references, also described methodology adequately with changes to support the rationale for our study.

If there are any further changes or clarifications you would like to suggest, we would be more than happy to implement them. We look forward to hearing your feedback and hope for a positive response.

Once again, we thank you for your time and effort in reviewing our manuscript.

Major comments:

  1. The authors need to clarify the relationship between phytoconstituents and the protein targets explored in the manuscript.

Response 1: Thank you for your valuable feedback. We have updated the discussion section of our manuscript to include information about the relationship between the phytoconstituents of Solanum xanthocarpum and Plumbago zeylanica and the protein targets explored.

  1. The authors need to analyze the angiogenesis pathway with more protein targets mapped.

Response 2: We have analyzed the information obtained from KEGG, mapped it to the angiogenesis pathway, and explained the relationship and pharmacological mechanisms of additional target proteins in relation to the EGFR tyrosine kinase inhibitor resistance pathway and diabetic retinopathy.

Minor comments:

  1. The authors need to summarize key findings of the story in the abstract, the current version looks like methodology.

Response 1: We have updated the entire abstract section to summarize the key findings and revised it to meet the journal's requirements.

  1. The methodology needs to be written in very concisely.

Response 2: We have made changes in the procedure of methodology and concise it.

  1. The acronyms should be expanded in the first instance.

Response 3: The acronyms were input in the first instance and have done all the corrections needed. Thank you for noticing.

  1. The authors need to include ligand efficiency in Table 1.

Response 4: We have included the formula for calculating ligand efficiency in the methodology section and added the ligand efficiency values to Table 3 into the bracket with binding energy.

  1. The authors need to focus more on the oxidative stress pathway with more proteins targeted.

Response 5: We have expanded the discussion to include additional protein targets involved in the oxidative stress pathway, such as the PI3K/Akt signaling pathway, and their pharmacological effects on the disease scenario and compound effects.

  1. The authors calculated pharmacodynamics in the manuscript, and they can also comment on pharmacokinetic properties.

Response 6: The pharmacokinetic properties of compounds and their related information have been included in Table 1 as per the suggestion. 

  1. The authors included several main figures, a few figures can be moved to supplementary.

Response 7: As per the reviewer’s suggestion, some of the figures have been moved to supplementary file, which can be found in the supplementary file.

  1. In the introduction, the authors should write more rationale for selecting Solanum xanthocarpum (SX) and Plumbago zeylanica(PZ) plants for diabetic retinopathy (DR).

Response 8: We have enhanced the rationale for selecting SX and PZ for diabetic retinopathy in the introduction section.

Reviewer 2 Report

Comments and Suggestions for Authors

In this manuscript, the authors conducted an iterative study for identifying the key compounds in Solanum xanthocarpum and Plumbago zeylanica in diabetic retinopathy (DR) treatment and their binding sites with DR target. Docking analysis and molecular dynamics (MD) simulations are employed to investigate the target-ligand binding and interactions. However, the manuscript, in its present form, exhibits significant weaknesses that necessitate substantial revisions. I outline some major issues that must be addressed through substantial revisions before reconsidering for publication.

1. Overall, the manuscript primarily consists of experimental reports, and the presentation of the results is lengthy and difficult to follow. Much of the information could be relegated to the supporting information. For instance, in the Materials and Methods section, is it necessary to include all details of the screening analysis in the main text? A brief summary would be more appropriate, with detailed information moved to the supplementary information, which is not available in current submission.

2. In the methods section, numerous essential citations related to MD simulations are missing, including references for GROMACS, CHARMM, water models, and ion parameters, etc. The authors are advised to consult and reference scientific journals appropriately. The omission of these references suggests a lack of familiarity with MD simulations, casting doubt on the reliability of the analysis regarding MD trajectories.

3. Concerning point 2, I recommend that the PDB files of the starting structures used in each simulation be provided in the supplementary information. This will allow for greater transparency and reproducibility of the results.

4. Now that MD simulations with explicit treatment to solvents and ions have been done, it becomes straightforward to further evaluate the ligand-targert interaction of corresponding systems. Lack of dynamic information, docking scores may not accurately reflect ligand binding interactions. Computational methods like MMPBSA could be utilized to provide a more comprehensive analysis of these interactions.

5. For the PCA analysis, instead of merely reporting the distributions of principal components, I recommend visualizing the motion of the target relative to ligands to determine what motions PC1 and PC2 represent. This can be easily accomplished through trajectory analysis and will be very helpful in aiding readers to understand the molecular mechanisms underlying the process.

6. The quality of the figures is low and appears stretched. Furthermore, the figure captions are not helpful in facilitating understanding. For example, it seems that the thickness of the arrows in Figure 3 represents interaction strength, but there is no explanation provided in either the text or the captions. Captions should assist readers in understanding the content depicted in the figures. This issue persists across almost all figures in the manuscript.

7. When defining any thresholds or cutoffs, references should be provided. For example, the definitions of BS and DL on page 5, and the definitions of definite interactions, strong binding activities, and good binding activity on page 15 require citations. Please review the manuscript thoroughly to ensure all such definitions are properly referenced.

8. All abbreviations should be defined the first time they appear in the text.

Author Response

Dear Reviewer,

We would like to sincerely thank you for taking the time to review our manuscript submitted to Biology MDPI. We greatly appreciate your valuable comments and suggestions, which have helped us to improve the quality of our work.

We are pleased to inform you that we have addressed all of your comments thoroughly. In particular, we have provided a more comprehensive background in the introduction, improved clarity of result section and included all relevant references, also described methodology adequately with changes to support the rationale for our study.

If there are any further changes or clarifications you would like to suggest, we would be more than happy to implement them. We look forward to hearing your feedback and hope for a positive response.

Once again, we thank you for your time and effort in reviewing our manuscript.

  1. Overall, the manuscript primarily consists of experimental reports, and the presentation of the results is lengthy and difficult to follow. Much of the information could be relegated to the supporting information. For instance, in the Materials and Methods section, is it necessary to include all details of the screening analysis in the main text? A brief summary would be more appropriate, with detailed information moved to the supplementary information, which is not available in current submission.

 Response 1: All the authors are thankful towards the reviewer for valuable insights. We have made changes throughout the manuscript, separated some redundant figures and provided through the supplementary file and also concise the methodology section by removing redundant information.

  1. In the methods section, numerous essential citations related to MD simulations are missing, including references for GROMACS, CHARMM, water models, and ion parameters, etc. The authors are advised to consult and reference scientific journals appropriately. The omission of these references suggests a lack of familiarity with MD simulations, casting doubt on the reliability of the analysis regarding MD trajectories.

 Response 2: As suggested by the reviewer, we have added the required references.

Van Der Spoel, D.; Lindahl, E.; Hess, B.; Groenhof, G.; Mark, A.E.; Berendsen, H.J.C. GROMACS: Fast, Flexible, and Free. Journal of Computational Chemistry 2005, 26, 1701–1718, doi:10.1002/jcc.20291.

Huang, J.; Rauscher, S.; Nawrocki, G.; Ran, T.; Feig, M.; De Groot, B.L.; Grubmüller, H.; MacKerell, A.D. CHARMM36m: An Improved Force Field for Folded and Intrinsically Disordered Proteins. Nature Methods 2016, 14, 71–73, doi:10.1038/nmeth.4067.

Price, D.J.; Brooks, C.L. A Modified TIP3P Water Potential for Simulation with Ewald Summation. Journal of Chemical Physics 2004, 121, 10096–10103, doi:10.1063/1.1808117.

Typing, A. of the C.G.F.F. (CGenFF) I.B.P. and A. CHARMM General Force Field (CG. Journal of Computational Chemistry 2010, 31, 671–690, doi:10.1002/jcc.21367.CHARMM.

Applequist, J.; Carl, J.R.; Fung, K.K. An Atom Dipole Interaction Model for Molecular Polarizability. Application to Polyatomic Molecules and Determination of Atom Polarizabilities. Journal of the American Chemical Society 1972, 94, 2952–2960, doi:10.1021/ja00764a010.

Sangster, M.J.L.; Atwood, R.M. Interionic Potentials for Alkali Halides. II. Completely Crystal Independent Specification of Born-Mayer Potentials. Journal of Physics C: Solid State Physics 1978, 11, 1541–1555, doi:10.1088/0022-3719/11/8/015.

  1. Concerning point 2, I recommend that the PDB files of the starting structures used in each simulation be provided in the supplementary information. This will allow for greater transparency and reproducibility of the results.

 Response 3: The PDB files of the starting structures of SRC (ID: 2BDJ) and HSP90AA1 (ID: 4BQG) are provided in the supplementary information.

  1. Now that MD simulations with explicit treatment to solvents and ions have been done, it becomes straightforward to further evaluate the ligand-targert interaction of corresponding systems. Lack of dynamic information, docking scores may not accurately reflect ligand binding interactions. Computational methods like MMPBSA could be utilized to provide a more comprehensive analysis of these interactions.

Response 4: In the light of the reviewer’s comment, we have performed MM-GBSA for better understanding of the protein-ligand interaction. We chose MM-GBSA over MM-PBSA due to limitation of the computational resources and time required to submit the revision. The method of MM-GBSA is given in section 2.10, and the results and discussion are given in section 3.10.

  1. For the PCA analysis, instead of merely reporting the distributions of principal components, I recommend visualizing the motion of the target relative to ligands to determine what motions PC1 and PC2 represent. This can be easily accomplished through trajectory analysis and will be very helpful in aiding readers to understand the molecular mechanisms underlying the process.

Response 5: Thank you for your valuable suggestion regarding the PCA analysis. While visualizing the motion of the target relative to the ligands could provide additional insights into the specific motions represented by PC1 and PC2, the primary objective of our current study was to assess the overall distributions and contributions of these principal components to the dynamics of the protein-ligand complexes.

We chose to focus on the distribution of principal components to highlight the dominant motions influencing the system's dynamics. While trajectory analysis is indeed a useful tool, it requires extensive computational resources and may fall outside the current scope of our study. However, we agree that incorporating such visualizations could enhance understanding, and we will consider this approach for future work to provide a more comprehensive view of the molecular mechanisms.

  1. The quality of the figures is low and appears stretched. Furthermore, the figure captions are not helpful in facilitating understanding. For example, it seems that the thickness of the arrows in Figure 3 represents interaction strength, but there is no explanation provided in either the text or the captions. Captions should assist readers in understanding the content depicted in the figures. This issue persists across almost all figures in the manuscript.

 Response 6: We have done correction as per reviewer has suggested for all the figures and thankful for noticing such mistake.

  1. When defining any thresholds or cutoffs, references should be provided. For example, the definitions of BS and DL on page 5, and the definitions of definite interactions, strong binding activities, and good binding activity on page 15 require citations. Please review the manuscript thoroughly to ensure all such definitions are properly referenced.

 Response 7: We have reviewed the entire manuscript and included all the required citations as requested by the reviewer.

  1. All abbreviations should be defined the first time they appear in the text.

Response 8: The abbreviations were defined in the first time they are appearing and have done all the corrections needed. Thank you for noticing.

Round 2

Reviewer 1 Report

Comments and Suggestions for Authors

I have no suggestions for this revised manuscript.  

Author Response

Dear Reviewer,

We would like to sincerely thank you for taking the time to review our manuscript submitted to Biology MDPI. We greatly appreciate your valuable comments and suggestions, which have helped us improve the quality of our work. 

We are happy to let you know that we have taken care of all of your comments thoroughly. 

If there are any further changes or clarifications you would like to suggest, we would be more than happy to implement them. We look forward to hearing your feedback and hope for a positive response.

Once again, we thank you for your time and effort in reviewing our manuscript.

Reviewer 2 Report

Comments and Suggestions for Authors

Most of my concerns regarding the manuscript have been addressed. I have no objections to its publication, provided that the remaining minor issues are resolved.

1. In page 5, 'solvation was accomplished with TIP3P water molecules via the pdb2gmx command in GROMACS.'

pdb2gmx doesn't do solvation.

2. In page 5, 'Sodium chloride (NaCl) was added to a concentration of 150 mM to replicate physiological conditions'

What are the parameter sets for Na+ and Cl- ions?

3. References still missing, e.g. Leap-frog, LINCS, etc.

4. The PDB files representing the starting points of MD simulations, i.e., the docking results, should be provided in the SI.

Author Response

Dear Reviewer,

We would like to sincerely thank you for taking the time to review our manuscript submitted to Biology MDPI. We greatly appreciate your valuable comments and suggestions, which have helped us improve the quality of our work. We are very thankful to you for giving us the chance to rectify the manuscript again.

We are happy to let you know that we have taken care of all of your comments thoroughly. 

If there are any further changes or clarifications you would like to suggest, we would be more than happy to implement them. We look forward to hearing your feedback and hope for a positive response.

Once again, we thank you for your time and effort in reviewing our manuscript.

Comment 1: 

In page 5, 'solvation was accomplished with TIP3P water molecules via the pdb2gmx command in GROMACS.'

pdb2gmx doesn't do solvation.

Response 1: 

The proper command to solvate i.e. ‘gmx solvate’ is corrected in the revised manuscript.

Comment 2: In page 5, 'Sodium chloride (NaCl) was added to a concentration of 150 mM to replicate physiological conditions'

What are the parameter sets for Na+ and Cl- ions?

Response 2: 

The following parameters were used, which we have updated in the manuscript as well:

The parameters for Na+ were: charge = +1.0 e, Lennard-Jones parameters: σ = 2.58 Å, and ε = 0.4184 kJ/mol. For Cl-, the parameters were: charge = -1.0 e, Lennard-Jones parameters: σ = 4.40 Å, and ε = 0.4184 kJ/mol. 

Comment 3: References still missing, e.g. Leap-frog, LINCS, etc.

Response 3: 

The references of Leap-frog integrator [41] and LINCS algorithm [42] are provided, as given in GROMACS manual.

Hockney, R.W.; Goel, S.P.; Eastwood, J.W. Quiet High-Resolution Computer Models of a Plasma. Journal of Computational Physics 1974, 14, 148–158, doi:10.1016/0021-9991(74)90010-2.

Hess, B.; Bekker, H.; Berendsen, H.J.C.; Fraaije, J.G.E.M. LINCS: A Linear Constraint Solver for Molecular Simulations. Journal of Computational Chemistry 1997, 18, 1463–1472, doi:10.1002/(SICI)1096-987X(199709)18:12<1463::AID-JCC4>3.0.CO;2-H.

Comment 4: The PDB files representing the starting points of MD simulations, i.e., the docking results, should be provided in the SI.

Response: The PDB files used at the starting point of MD simulations are provided in the Supplementary file, which is named "SRC-Apigenin docked complex" and "HSP90AA1- Isozeylanone docked complex".